# Enhanced Lung Cancer Detection Using a Combined Ratio of Antigen–Autoantibody Immune Complexes against CYFRA 21-1 and p53

**DOI:** 10.3390/cancers16152661

**Published:** 2024-07-26

**Authors:** Heyjin Kim, Jin Kyung Lee, Hye-Ryoun Kim, Young Jun Hong

**Affiliations:** 1Department of Laboratory Medicine, Korea Cancer Center Hospital, Korea Institute of Radiological and Medical Sciences, Seoul 01812, Republic of Korea; heyjin@kirams.re.kr (H.K.); jklee@kirams.re.kr (J.K.L.); 2Division of Pulmonology, Department of Internal Medicine, Korea Cancer Center Hospital, Korea Institute of Radiological and Medical Sciences, Seoul 01812, Republic of Korea; slowly7@kirams.re.kr

**Keywords:** lung cancer, autoantibody, biomarker, CYFRA 21-1, p53

## Abstract

**Simple Summary:**

This study investigated a new blood test (9G test^TM^ Cancer/Lung) for early lung cancer detection. This test measures complexes of autoantibodies with specific lung cancer antigens for CYFRA 21-1 and p53. This study found that the diagnostic performances were significantly better than traditional methods at identifying LC, especially in early stages. The test showed good overall accuracy (AUC of 0.945) and detected early-stage lung cancer with high sensitivity (87.5%). This suggests it could be a valuable tool alongside existing scans like X-rays and LDCT for improving early LC detection.

**Abstract:**

The early detection of lung cancer (LC) improves patient outcomes, but current methods have limitations. Autoantibodies against tumor-associated antigens have potential as early biomarkers. This study evaluated the 9G test^TM^ Cancer/Lung, measuring circulating complexes of two antigen–autoantibody immune complexes (AIC) against their respective free antigens (CYFRA 21-1 and p53) for LC diagnosis. We analyzed 100 LC patients and 119 healthy controls using the 9G test^TM^ Cancer/Lung, quantifying the levels of AICs (CYFRA 21-1-Anti-CYFRA 21-1 autoantibody immune complex (CIC) and p53-Anti-p53 autoantibody immune complex (PIC)), free antigens (CYFRA 21-1 and p53), and ratios of AICs/antigens (LC index). The levels of the CICs and PICs were significantly elevated in LC compared to the controls (*p* < 0.0062 and *p* < 0.0026), while free antigens showed no significant difference. The CIC/CYFRA 21-1 and PIC/p53 ratios were also significantly higher in LC (all, *p* < 0.0001). The LC index, when combining both ratios, exhibited the best diagnostic performance with an area under the curve (AUC) of 0.945, exceeding individual CICs, PICs, and free antigens (AUCs ≤ 0.887). At a cut-off of 3.60, the LC index achieved 81% sensitivity and 95% specificity for LC diagnosis. It detected early-stage (Stage I–II) LC with 87.5% sensitivity, exceeding its performance in advanced stages (72.7%). The LC index showed no significant differences based on age, gender, smoking status (former, current, or never smoker), or pack years smoked. The LC index demonstrates promising potential for early LC diagnosis, exceeding conventional free antigen markers.

## 1. Introduction

Lung cancer (LC) remains a leading cause of global mortality, impacting both men and women with a concerningly low survival rate compared to other cancers [1]. Early detection is crucial for successful treatment, and recent advancements in screening and therapeutic strategies have played a significant role in reducing mortality. A recent report in Korea observed 5-year relative survival rates of 82% and 59% in Stage I and II non-small cell lung cancer (NSCLC) patients, respectively, while rates of 16% and 10% were observed in Stage III and IV, respectively [2]. Low-dose chest computed tomography (LDCT) screening has shown promise in high-risk individuals with a smoking history by increasing the detection of early-stage, potentially curable LC, leading to potentially reduced mortality rates [3]. This success has fueled global interest in identifying high-risk populations and implementing LDCT screening programs [4]. While LDCT offers benefits, it can generate false positives, leading to unnecessary procedures and costs [3,5]. Understanding the underlying mechanisms of LC is crucial for developing improved diagnostic tools and treatment strategies.

LC is a multifaceted disease arising from a complex interplay of genetic predisposition, environmental exposures (including carcinogens from smoking and other sources that damage lung cell DNA), and immune dysfunction [6]. These factors contribute to a breakdown of cellular and molecular processes. At the genomic level, mutations like oncogene activation, tumor suppressor gene inactivation, or epigenetic alterations are believed to play a role in LC development [7]. In addition to genetic factors, metabolic factors such as glucose transporters, as well as other cell cycle regulators, also contribute to the development of LC and the differentiation of tumor cells [8].

The ongoing search for an ideal LC biomarker has led researchers to explore various protein markers. Tumor cells can express unique or overexpressed proteins (tumor-associated antigens (TAAs)) that the immune system recognizes as foreign, triggering an immune response with autoantibody production [9]. Additionally, the inflammatory nature of the tumor microenvironment can further stimulate the immune system and contribute to autoantibody production [9]. Some autoantibodies may appear in the blood before traditional diagnostic methods can detect cancer, offering a potential window for earlier diagnosis and intervention [8]. This rationale stems from the observation that, during early-stage cancer, tumor antigens often exhibit a slow rise in concentration with a relatively small amount, while autoantibody levels against them increase rapidly with large quantities [10,11].

Several TAAs, including NY-ESO-1, MUC1, p53, TNF-α, CEA, CYFRA 21-1, NSE, and CA125, have been identified as potential LC markers [12]. Among these promising TAAs, Cytokeratin 19 (CK19) stands out as a protein highly expressed in LC and a structural component of the bronchi’s simple epithelium [13]. CYFRA 21-1, a fragment of CK19, has been associated with LC staging, with its positive rate and level increasing with disease progression and poorer prognoses [13,14,15]. In advanced NSCLC patients, high CYFRA21-1 levels are associated with shorter progression-free survival and overall survival [13,16]. Mutations in the p53 tumor suppressor gene are prevalent in various malignancies, including LC [17]. Mutations in the p53 gene play a crucial role in LC progression and prognosis. Studies have shown that mutated p53 accumulation in LC is associated with a significantly negative prognostic value, indicating a more aggressive disease course [18,19,20]. The mutated p53 gene accumulates in tumor cells, and a specific humoral immune response can be mounted against these mutant p53 proteins, resulting in p53 autoantibody production [17,21,22]. p53 autoantibodies are linked to poorer survival outcomes in LC [23]. p53 autoantibodies are rarely detected in healthy controls (HCs) or those with non-neoplastic diseases, suggesting their potential as a tumor marker [24].

While TAAs hold promise, they often lack sufficient sensitivity or specificity for a definitive diagnosis on their own [12,25]. Studies exploring panels targeting various TAAs have demonstrated improved sensitivity compared to single markers [25]. However, autoantibody tests are not yet widely used in routine LC diagnosis, primarily due to the limitations in sensitivity and specificity associated with specific detection methods. The 9G DNA chip, a lateral flow microarray platform developed by Biomatrix Technology, utilizes DNA oligonucleotide–guanine capture technology to enable the detection of tumor markers at concentrations as low as 1 pg/mL [26]. Given the potential for autoantibodies to form complexes with their cognate antigens, the ratio of the antigen–autoantibody immune complex (AIC) to target antigens may offer a means to discriminate between HCs and cancer patients (Figure 1). In our previous studies, we evaluated a 9G DNA chip-based autoantibody detection method to identify the AICs and antigens against each CYFRA 21-1 and p53 in LC patients, demonstrating their potential usefulness in LC diagnosis with high sensitivity and specificity among other candidate biomarkers [27,28,29,30]. This study aimed to assess the diagnostic utility of autoantibody–antigen complexes against CYFRA 21-1 and p53, along with their ratio combinations, for LC diagnosis.

## 2. Materials and Methods

### 2.1. Study Participants

Clinical samples derived from patients diagnosed with LC (*n* = 100) and the HCs (*n* = 119) were collected and tested in 2020 at the Korea Cancer Central Hospital. We retrospectively reviewed the patients’ electronic medical records through an honest broker. The medical records included patient age, sex, smoking history, evaluation methods for pathologic diagnosis, pathologic type, the lung mass size on LDCT exam, and clinical stage. LC patients included Stages I–IV for NSCLC and limited (LD) to extensive disease (ED) for small cell lung cancer (SCLC), as determined by the eighth edition of the TNM staging system and two-stage system, respectively (Table 1 and Appendix A) [31,32]. All LC patients underwent chest radiography and LDCT screening: patients who had abnormal findings on chest X-ray (CXR) underwent LDCT. Biopsies were then performed on patients with LDCT findings suggestive of malignancy, such as lung masses, or other abnormalities including nodules, linear branching, patchy airspace changes, nodular forms suspected of tuberculosis, multiple small ground-glass opacities, or ill-defined lesions suspected of being benign tumors. Pathologic diagnosis was confirmed via paraffine-embedded tissues analysis after surgery (*n* = 53) or percutaneous needle biopsy (*n* = 47). Pathologically, NSCLC patients not diagnosed with a subtype of squamous cell carcinoma or adenocarcinoma were classified as “other types” (Table 1). Blood samples from the patients with LC were collected before treatment, such as via surgery, chemotherapy, and radiotherapy. HCs were excluded if they reported a history of cancer or infectious diseases, as well as if they produced abnormal findings on questionnaires, CXR, or the complete blood count (CBC) during their annual health checkup. K2 ethylenediaminetetraacetic acid (EDTA)-anticoagulated remnant blood specimens were collected from all participants after a CBC test. All samples were stored at 4 °C before centrifugation for 10 min at 2000× *g* at 4 °C. The resulting plasma was archived at −70 °C in the Korea Institute of Radiological and Medical Sciences (KIRAMS) Radiation Biobank (KRB-2020-I007). We selected four markers of two proteins (CYFRA 21-1 and P53) that have been reported to be useful in the detection of LC in previous studies [28,29,30,31,32,33]. Four markers were measured in the plasma samples from all participants: CYFRA 21-1-anti-CYFRA 21-1 AIC (CIC), CYFRA 21-1 antigen, p53-anti-p53 AIC (PIC), and p53 antigen. This study was approved by the Institutional Review Board of the Korea Institute of Radiological and Medical Sciences (KIRAMS) (IRB#-2020-06-017-002).

### 2.2. Methods

The general procedures for the detection of the four markers of the 9G test^TM^ Cancer/Lung (Biometrix Technology Inc., Chuncheon, Republic of Korea) were described in detail in previous reports [29,30]. Here, the procedures will be described in summary. Each marker for CYFRA 21-1 and p53 shows the process of these two methods side by side. The procedure for CIC or PIC detection in plasma samples is described as follows:Preparation—A plasma sample (20 µL) is mixed with the detection mixture (containing CYFRA 21-1-cAb-DNA (p53-cAb-DNA) and anti-human-IgG-FB).Incubation—The mixture is incubated to allow for the formation of the CIC or PIC complexes.Loading—The mixture is loaded onto a 9G DNA membrane.Hybridization and washing—The membrane is hybridized with a probe specific for CIC or PIC, followed by washing to remove unbound probes.Scanning—The membrane is scanned using a fluorescence scanner to detect the presence of the CIC complex.

The procedure for the antigen (CYFRA 21-1 or p53) in plasma samples is described as follows:Preparation—A plasma sample (20 µL) is mixed with the detection mixture (containing CYFRA 21-1-dAb-FB or p53-dAb-FB).Incubation—The mixture is incubated to allow for the formation of the CYFRA 21-1-dAb-FB or p53-dAB-FB complexes.Loading—The mixture is loaded onto a 9G DNA membrane.Hybridization and washing—The membrane is hybridized with a probe specific for CYFRA 21-1 or p53, followed by washing to remove unbound probes.Scanning—The membrane is scanned using a fluorescence scanner to detect the presence of CYFRA 21-1 or p53.

In previous method research, the following equation represents a lung cancer index [27,29]. We brought the equation and adjusted it with clinical samples. Finally, the combination ratios for two to four markers, including CIC/CYFRA 21-1 and PIC/p53, were determined according to Equation (1), and they were then used to discriminate between HCs and patients with LC.

This is the equation for the LC index:LC Index = [(CIC/CYFRA 21-1) × (PIC/p53)].(1)

The LOD for p53 and PIC detection were 7.41 pg/mL and 5.74 pg/mL, respectively. The detection ranges for both biomarkers were 50–7500 pg/mL. For both CYFRA 21-1 and CIC, the LOD were 0.03 ng/mL, and the detection ranges were 0.03–10.00 ng/mL. Sensitivity and specificity were determined using cut-off levels to distinguish between the HCs and LC patients. The cut-off levels were aimed at the area under the curve (AUC).

The following markers were measured in the plasma samples from all participants: CIC, CYFRA 21-1, PIC, and p53. This study was approved by the Institutional Review Board of the Korea Institute of Radiological and Medical Sciences (KIRAMS) (IRB#-2020-06-017-002).

### 2.3. Statistical Analysis

A normality test was performed using the Shapiro–Wilk test. Non-normality distributed continuous data are presented as medians with interquartile ranges (IQR), while categorical data were presented as counts and percentages. The Chi-squared test or Mann–Whitney U test was used to compare two groups (e.g., HCs vs. LC patients). For comparisons between three or more groups on a continuous variable, the Kruskal–Wallis test was employed. All analyses were two-sided with *p*-values of <0.05 considered statistically significant. The receiver operating characteristic (ROC) curve was used to assess the overall diagnostic performance. The optimal cut-off point was determined using the closest top-left method on the ROC curve. Sensitivity, specificity, accuracy, the positive predictive value (PPV), and negative predictive value (NPV) were calculated at 95% confidence intervals (CI). Statistical analyses were performed using Analyze-it version 5.90 (Analyze-it Software Ltd., Leeds, UK) and Rex-Pro version 3.6.1.0 (RexSoft Inc., Seoul, Republic of Korea).

## 3. Results

### 3.1. Diagnostic Performance

The levels of CIC and PIC effectively discriminated the LC patients from the HCs (*p* = 0.0062 and *p* = 0.0026, respectively) (Table 2). CIC/CYFRA 21-1, PIC/p53, and the LC index were all significantly higher in LC patients compared to the HCs (all, *p* < 0.0001).

The ROC analyses of the four individual markers, two ratios, and the LC index are presented in Figure 1. The LC index achieved 81.0% sensitivity and 95.0% specificity for diagnosing all LC stages (I–IV) at the optimal cut-off of 3.60 (Table 3). Notably, the LC index demonstrated greater sensitivity for early-stage LC (Stage I–II NSCLC and limited SCLC) compared to advanced stages (87.5% vs. 72.7% sensitivity at a fixed 95% specificity) (Table 4). This trend continued within the early-stage LC, particularly for adenocarcinoma of NSCLC (Table 5). The LC index (cut-off of 3.60) exhibited a higher sensitivity compared to the later stages when maintaining 95% specificity (Table 4). 

### 3.2. Utility as a Complementary Test to Radiologic Exams

The CXR findings agreed with the LDCT findings in 84 of 100 of LC patients. Higher concordance was observed for detecting small nodules or other abnormalities compared to lung masses on CXR, LDCT, and the combined CXR*LDCT approach (Table 6). Among patients with a small nodule or another abnormality identified on CXR, 88.2% (15 out of 17) had a positive LC index (cut-off of ≥3.60) on LDCT. Similarly, 90.9% (10 out of 11) of patients with these findings on LDCT also had a positive LC index.

### 3.3. LC Index Level According to the Basic Characteristics in Lung Cancer Patients and Healthy Controls

The LC index level in LC patients and HCs was evaluated according to various demographic and smoking history factors (Table 7). No significant differences were observed in the LC index levels between patients younger and older than 60 years, males and females, or across the three smoking categories (former, current, and never smoker). Additionally, no significant associations were found between the LC index and 20 pack years smoked in former and current smokers (*p* > 0.05).

## 4. Discussion

The discovery of autoantibodies’ potential for early cancer diagnosis has spurred the development of numerous protein detection methods. Autoantibodies are produced in response to tumor-associated antigens. The identification of numerous tumor-associated proteins and a multitude of studies employing diverse detection methodologies can be detected in peripheral blood, offering a minimally invasive, presymptomatic, and low-cost approach for screening and prognosis classification. However, accurately detecting the autoantibodies of these proteins in blood remains a significant technical challenge [33]. Consequently, widespread adoption in clinical laboratories has not yet materialized. This limited adoption can be attributed to the low sensitivity and specificity demonstrated in early-stage cancers, particularly LC, which carries a poor prognosis [25]. In this study, we assessed the diagnostic potential of the 9G test™ Cancer/Lung for LC. The test targets CYFRA 21-1 and p53 demonstrated promising results with high overall sensitivity (81%) and specificity (95%). Notably, the assay achieved an even greater sensitivity (87.5%) for early-stage LC while maintaining the same high specificity (95%).

The 9G test utilizes a novel concept by exploiting high-affinity AICs against target antigens [26,30,34]. This method enables the detection of both the total CYFRA 21-1 or p53 protein (encompassing wild-type and mutant forms) and the corresponding AICs (CIC and PIC), potentially indicating cancer presence [29,30]. This approach offers superior sensitivity and specificity in differentiating LC patients from HCs. Compared to traditional protein microarrays, the 9G test™ facilitates enhanced protein stability and affinity, enabling the detection of AICs or antigens at low-concentration without amplification steps. This contributes to improved sensitivity, which is particularly beneficial for early-stage LC detection.

A signature of multiple autoantibodies likely offers a higher diagnostic value than a single marker. A recent systematic review and meta-analysis assessed the EarlyCDT^®^-Lung Test, the most-studied panel targeting seven tumor-associated autoantibodies against p53, NY-ESO-1, CAGE, GBU4-5, HuD, MAGE-A4, and SOX2. Despite promising aspects, the test demonstrated a detection rate (sensitivity) of 47% for LC across all stages, with a high specificity of 90% using enzyme-linked immunosorbent assay (ELISA) [25]. Our previous study using the 9G DNA chip method showed better performance with the combinations of ratios derived from four protein markers (CYFRA 21-1, ProGRP, NSE, and NGAL) compared to individual marker ratios [28]. However, using multiple markers increases testing time, cost, and requires additional quality control processes in clinical settings. Therefore, we focused on ratios against the two proteins, CYFRA 21-1 and p53, which exhibited the best diagnostic performance in our previous studies [28,29,30]. This study found better performance in markers targeting p53 compared to CYFRA 21-1, even though p53 is a common target in various cancers, including breast, colorectal, and LC [17,35,36,37,38]. The p53 protein plays a critical role in suppressing tumor growth, and mutations or abnormalities in its function are associated with LC. Mutations in p53 occur in 50% to 90% of LC patients [20,35]. CYFRA 21-1 is also a well-established protein marker for diagnosis and prognosis of NSCLC [13,39,40]. Its diagnostic efficacy is positively correlated with cancer stage and has a higher prevalence in squamous cell carcinoma among NSCLC subtypes [39,40].

Smoking history, a well-established LC risk factor, significantly contributes to the disease process [8]. It does so by inducing chronic inflammation in lung cells, causing mutations in genes controlling cell division and growth, promoting uncontrolled cancer cell proliferation, and weakening the immune system’s ability to identify and eliminate abnormal cells [8,20,41]. This allows mutated cells to evade immune defenses and fuel tumor growth [41]. Our study found a significantly higher frequency of smoking exposure in the LC group (*p* < 0.001). However, LC index levels did not differ significantly between smokers and non-smokers within the LC group. This suggests that smoking-related changes likely do not impact the LC index’s ability to differentiate LC. Additionally, no significant differences in LC index levels were observed based on gender or age, suggesting its potential as a gender- and age-independent diagnostic tool (Table 7).

While this study showed promising results, limitations include the retrospective design and a relatively small patient cohort. Consistent with the previous study, this study also demonstrated better performance in the adenocarcinoma type compared to squamous cell carcinoma or other types, achieving a sensitivity of 85.5% and a specificity of 95.0%. SCLC displayed the lowest sensitivity (70%) among all LC types, likely due to the selection of CYFRA 21-1, which is more specific for NSCLC; meanwhile, p53, a marker present across various cancers, is more commonly detected in SCLC patients [20,35,39,40,42]. The LC index yielded a much higher sensitivity of 91.7% in Stage II NSCLC compared to the previous results [27,28]. However, sensitivity declined in advanced stages (Table 4). LC index levels in advanced stages (NSCLC Stage IV and SCLC extensive disease) showed a non-significant downward trend (all *p* > 0.05, Appendix A). These findings align with the prior research on autoantibody characteristics, suggesting that TAAs may undergo alterations or become less immunogenic as cancer progresses, leading to reduced autoantibody production [9,11]. Future studies with larger, prospective cohorts are warranted to confirm these findings across all LC stages and pathological types.

In addition, including samples from individuals with benign nodules would be valuable for assessing the LC index as a complementary tool to imaging diagnostics, which can have high false-positive rates. Lung nodules are small (less than 3 cm) abnormal areas detected on CXR or CT scans. Most lung nodules are benign, and they are often caused by old infections, scar tissue, or other factors. However, additional tests like invasive biopsies or follow-up CT scans and positron emission tomography (PET) scans are often needed to determine if a nodule is cancerous, causing significant psychological stress for patients. In our study, 17% and 11% of LC patients presented with small nodules or other ambiguous findings on initial CXR and LDCT scans, respectively. Additionally, 6 of the 84 patients with matching results on both CXR and LDCT also had these findings (Table 6). When the LC index results were considered alongside these radiological findings, 88.2% (15 of 17), 90.9% (10 of 11), and 83.3% (5 of 6) of patients were ultimately diagnosed with LC. While this study compared results with radiological data, it excluded the incidental lung nodules identified in the HCs. This exclusion limits the generalizability of the findings to the broader LC patient population. Further studies comparing malignant and benign nodules could improve the differentiation of LC from other abnormalities detected in radiological exams.

This study represents an initial step in validating the efficacy of this novel protein marker combination for diagnosing LC. Our findings are encouraging and support its promising diagnostic potential. When combined with LC patients’ imaging results, the LC index has the potential to provide stronger evidence for the presence of cancer. The 9G test™ Cancer/Lung could be particularly valuable in identifying patients with suspicious lung nodules detected by LDCT scans, potentially facilitating more targeted interventions. Future studies including samples from individuals with benign nodules are warranted to evaluate the LC index as a complementary tool to imaging diagnostics, which are known for their high false-positive rates. However, this evaluation hinges on confirming the test’s ability to differentiate LC from non-cancerous conditions.

## 5. Conclusions

Our findings suggest that the ratios of CIC to CYFRA 21-1 and PIC to p53, measured by the 9G test™ Cancer/Lung, hold promise as a complementary tool for LC screening alongside radiological tests. This LC index, formed by these ratios, could potentially improve early detection and reduce the need for unnecessary biopsies. Further validation studies are needed to confirm the LC index’s effectiveness in differentiating LC from benign nodules and its potential for monitoring recurrence during treatment. Additionally, optimizing the 9G test process for wider clinical application and cost-effectiveness is crucial for its real-world implementation.

## Data Availability

The data presented in this study are available on request from the corresponding authors.

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
