# Peer review of "Enhanced Lung Cancer Detection Using a Combined Ratio of Antigen–Autoantibody Immune Complexes against CYFRA 21-1 and p53"

_cancers, 2024, doi:10.3390/cancers16152661_

Round 1

Reviewer 1 Report

Comments and Suggestions for Authors

1. No information is provided about the stage of lung cancer, growth form (central, peripheral), or damage to the lymph nodes. Add this information to Table 1. Further in the text of the manuscript, the stages are given, but they must be given in the description of the group too. There is no information about the presence of concomitant inflammatory pathology; this may affect the results.

2. It is not clear why calculate the level of CIC and PIC and free antigens on the combined group (section 3.1)?

3. Figure 2 repeats the data in Table 2; is it necessary to show both?

4. Add diagnostic accuracy in % to Table 5, this will be more informative.

5. Why is diagnostic sensitivity lower in advanced stages of lung cancer than in early stages?

Author Response

Response to Reviewer 1.

Thank you for your valuable comment. We've mostly tried to rewrite it to reflect your comments. We've written below in response to individual comments. We've highlighted new additions in yellow and We've changed the structure of the sentences for better readability in red color. 

Comment 1: No information is provided about the stage of lung cancer, growth form (central, peripheral), or damage to the lymph nodes. Add this information to Table 1. Further in the text of the manuscript, the stages are given, but they must be given in the description of the group too. There is no information about the presence of concomitant inflammatory pathology; this may affect the results.

Response 1: We have added the following content to the Study Participants section of the Methods to address the points you raised. We have added a description of the stages of lung cancer (line 112-115) with further specific TNM stages of NSCLC in Table S1 and Figure S1 to distinguish the status of lymph node tumor invasion. In addition, we added an analysis of the LC index level between lung cancer stages. We confirmed that there was no difference between the LC index level and the stage within LC patients. According to the recommendations of IASLC or AJCC, TNM staging is recommended for SCLC patients, but due to the limitations of the retrospective study, the clinical staging at diagnosis was divided into LD and ED, and it was difficult to determine the additional TNM stage. It was also difficult to obtain additional diagnoses for inflammatory pathological findings in all patients.

Comment 2: It is not clear why calculate the level of CIC and PIC and free antigens on the combined group (section 3.1)?

Response 2: Most immunological methods for the detection of each lung cancer-associated protein detect an antigen or autoantibody. Therefore, each step included a review of the method development process, but in this study, as you pointed out, it was not considered necessary, so we removed it for 3.1.

Comment 3: Figure 2 repeats the data in Table 2; is it necessary to show both?

Response 3: We have decided to keep only one table, Table 2, as it is easier to see the information about the figures and have removed Figure 2 as it is redundant.

Comment 4: Add diagnostic accuracy in % to Table 5, this will be more informative.

Response 4: As the patients included in the analysis were all pathologically proven LC patients, we separated the diagnosis of lung mass from other abnormal findings in each radiological method and analyzed the accuracy of the LC index (+)/(-) results for each radiological method and presented them in Table 6 (previously Table 5). As the patients included in the analysis were all pathologically proven LC patients, we separated the diagnosis of lung mass from other abnormal findings in each radiological method and analysed the accuracy of the LC index (+)/(-) results for each radiological method and presented them in Table 6 (previously Table 5).

Comment 5: Why is diagnostic sensitivity lower in advanced stages of lung cancer than in early stages?

Response 5: In the discussion section, we added the mention the results of lower sensitivity in advanced stage than in early stage and the results of LC index level analysis according to the 8th TNM stage classification. We write the following based on previously reported studies on autoantibody production in cancer (line 320~325).

Reviewer 2 Report

Comments and Suggestions for Authors

please underline in the intriduction the role of CYFRA and p53 as  markers of disease progression

what staging edition of lung cancer do you refer to?

about HCs the meaning should be state. what kind of biopsy was performed?

how could the two populations be compared, for example smoking could be a common factor. what were the inclusion criteria? I think that healthy people could be considered without neoplasia but with pack-years of at least 20

In the methods it is not clear what the LDCT screening was based on. Did this concern the HC group of smokers?was there a specific stage in the LC  group who undergoes blood sampling?

in the general baseline data I would include pack-year, smoking status, comorbidities, since in about 30% of long-term smokers nodules or masses can be found radiologically.

I suggest to include some references for improving the discussion comparing CYFRA, p53 with other biomarkers of disease progression. I would also discuss the importance of smoking cessation in smokers at risk for lung cancer.

Thorac Cancer. 2020 Nov;11(11):3060-3070. 

Author Response

Thank you for your valuable comment. We've mostly tried to rewrite it to reflect your comments. We've written below in response to individual comments. We've highlighted new additions in yellow and We've changed the structure of the sentences for better readability in red color. 

Comment 1: Please underline in the intriduction the role of CYFRA and p53 as markers of disease progression.

Response 1: We have revised the Introduction to add information about the role of the two proteins (CYFRA21-1, p53) in lung cancer progression and their research (line 76-77, line 79-81, 84).

Comment 2: what staging edition of lung cancer do you refer to?

Response 2: We added information about the stage classification reference in the participant section of the materials and methods (line 110-115).

Comment 3: about HCs the meaning should be state.

Response 3: The description of HCs was incorrectly labelled as patients and the description of HCs has been added in the participant section of the materials and methods (line 125-127).

Comment 4: what kind of biopsy was performed?

Response 4: We have re-investigated the biopsy methods in pathological diagnosis and added it to the methods section (line 120-122).

Comment 5: how could the two populations be compared, for example smoking could be a common factor. what were the inclusion criteria? I think that healthy people could be considered without neoplasia but with pack-years of at least 20?

Response 5: The smoking history of lung cancer patients and HC was re-examined and added to Table 1 according to the comments. The difference in LC index values was also confirmed and added to Table 7.

Comment 6: In the methods it is not clear what the LDCT screening was based on. Did this concern the HC group of smokers? was there a specific stage in the LC group who undergoes blood sampling?

Response 6: Patients diagnosed with lung cancer at KCCH who have CXR abnormalities identified in primary care are referred for further evaluation (LDCT) as well as repeat CXR at our hospital, and if a mass is suspected, additional tests are performed based on the patient's symptoms and findings, including pathological examination by bronchoscopy and CBC (line 112-125)

Comment 7: in the general baseline data I would include pack-year, smoking status, comorbidities, since in about 30% of long-term smokers nodules or masses can be found radiologically.

Response 7: The smoking history of lung cancer patients and HC was re-examined and added to Table 1 according to the comments. The difference in LC index values was also confirmed and added to Table 7. We have confirmed that there are no other benign conditions and those associated with lung disease by reviewing the questionnaires and CXR. This has been corrected by adding a description in the Methods section.

Comment 8: I suggest to include some references for improving the discussion comparing CYFRA, p53 with other biomarkers of disease progression. I would also discuss the importance of smoking cessation in smokers at risk for lung cancer. Thorac Cancer. 2020 Nov;11(11):3060-3070.

Response 8: Based on the above-mentioned paper, we have added metabolic factors in the pathogenesis of lung cancer in the Introduction section (line 57-59) and the reference to the carcinogenesis of smoking in the Discussion section (line 301-305).

Reviewer 3 Report

Comments and Suggestions for Authors

The author should justify the error bars in figure (1A, B and 2A, B)

Author may use a comparative analysis with most commonly used diagnostic method of lung cancer detection and studied techniques. What are the technical advancement over the previous diagnostic methods.

Author Response

Response to Reviewer 3.

Thank you for your valuable comment. We've mostly tried to rewrite it to reflect your comments. We've written below in response to individual comments. We've highlighted new additions in yellow and We've changed the structure of the sentences for better readability in red color. 

Comment 1: The author should justify the error bars in figure (1A, B and 2A, B).

Response 1: In response to reviewer 1's comments, we removed figures 1 (A&B) and 2 (A&B), which were deemed unnecessary. Results section 3.1 was also removed. Figure 2 was removed, and Table 2 was retained.

Comment 2: Author may use a comparative analysis with most commonly used diagnostic method of lung cancer detection and studied techniques. What are the technical advancement over the previous diagnostic methods.

Response 2: The above comment is addressed in the discussion section(line 276-300). Comparisons with many existing methods can be confusing and there are many existing comparative papers, so we have referenced their results [ref.12, 25] and an improvement over the technique used in this paper is the high sensitivity based on the high specificity of the CYFRA21-1, p53 marker selection.

Round 2

Reviewer 1 Report

Comments and Suggestions for Authors

I have no further comments on the manuscript. I believe that in its present form the article can be recommended for publication.